# Associations between handedness and brain functional connectivity patterns in children

Dardo Tomasi [1] ✉ & Nora D. Volkow[1]

Handedness develops early in life, but the structural and functional brain connectivity patterns associated with it remains unknown. Here we investigate associations between handedness and the asymmetry of brain connectivity in 9- to 10-years old children from the Adolescent Brain Cognitive Development (ABCD) study. Compared to right-handers, left-handers had increased global functional connectivity density in the left-hand motor area and decreased it in the right-hand motor area. A connectivity-based index of handedness provided a sharper differentiation between right- and left-handers. The laterality of hand-motor connectivity varied as a function of handedness in unimodal sensorimotor cortices, heteromodal areas, and cerebellum ($P < 0.001$) and reproduced across all regions of interest in Discovery and Replication subsamples. Here we show a strong association between handedness and the laterality of the functional connectivity patterns in the absence of differences in structural connectivity, brain morphometrics, and cortical myelin between left, right, and mixed handed children.

Handedness, the preference for using one hand over the other, is a trait associated with complex brain asymmetries influenced by genetics, environment, and neurodevelopment[1,2]. Left-handedness has a prevalence of 9–10%[3] with some variations based on ancestry[4]. Even though hand dominance is evident at 6 months of age for the majority of children[5] and appears to be present even at 18 weeks of gestational age or earlier[6], few studies have evaluated its association with brain asymmetries and their neurodevelopmental trajectories. In addition to genetic factors, it has been suggested that handedness and brain asymmetries are driven in part by developmental exposures to lateral biases in caregiver behavior[7], though others have failed to confirm the influence of early environmental factors in handedness or brain asymmetries in adults[8]. A study on the development of brain asymmetries done in a large cohort of 6–10-year-old children reported both decreases and increases in laterality with aging as assessed with resting state functional connectivity[9]. Greater lateralization in regions of the visual network (left calcarine gyrus) and the default mode network (right superior medial gyrus and right precuneus) were associated with right-handedness. As such most studies on the effects of handedness on brain morphometry have been done in adults, and their results are inconclusive. Specifically, while one study documented lower rightward asymmetry of cortical thickness in frontoparietal areas for 10

non-dextral compared to 67 dextral healthy adults[10] another reported higher cortical thickness in the right auditory cortex for 32 non-dextral compared to 34 dextral healthy adults[11]. Large-scale studies done in adults that relied on brain atlases with limited spatial resolution have not found significant associations with handedness[12,13]. In contrast, a large study from the UK Biobank dataset found that left-handers had lower surface area asymmetry in the anterior insula, fusiform, anterior middle cingulate, and precentral cortices and reduced leftward thickness asymmetry along postcentral gyrus than adult right-handers[14]. Studies on white matter diffusion metrics are also inconclusive, with one study done in adults reporting lower fractional anisotropy in prefrontal and limbic regions for 40 left-handers than for 42 right-handers[15] whereas a study in children reported no differences in white matter microstructure between 2646 right- and 293 left-handers[16].

Brain activation studies have linked left-handedness with differential activation of the hand areas in the primary motor cortex (M1), anterior lobe of the cerebellum, intraparietal sulcus (IPS), and premotor and motor cortex during functional MRI (fMRI) while performing motor tasks[17–19]. Functional magnetic resonance imaging (fMRI) studies done in adults reported that right-handers ($n = 142$) deactivated the ipsilateral M1 when moving their non-dominant hand

---

[1]National Institute on Alcohol Abuse and Alcoholism, Bethesda, MD 20892, USA. ✉e-mail: dardo.tomasi@nih.gov

than their dominant hand, whereas in left-handers ($n = 142$), ipsilateral M1 deactivation tended to be similar when moving either hand[19]. Studies on functional connectivity and handedness have been limited, with one study reporting weaker interhemispheric connectivity between left M1 and right premotor area in adult left-handers ($n = 18$) than in right-handers ($n = 18$)[20]. Otherwise, the specific patterns of functional connectivity associated with handedness in general and their emergence during childhood are mostly unknown.

Here we took advantage of the adolescent brain cognitive development (ABCD) study[21] to investigate the effects of handedness on brain structure during childhood. The large and diverse sample from the ABCD study[21] also gave us the opportunity to investigate the reproducibility of the effects of handedness on brain structure and function. For this study, we selected a total of 1800 children, comprising 600 left-handed, 600 right-handed, and 600 mixed-handed individuals. The selection ensured matching for sex, age, race, scanner manufacturer, family income, head motion, and total brain volume across the three groups. This sample size was carefully chosen to facilitate accurate participant matching and to enable a robust assessment of reproducibility. This is relevant since most brain imaging studies on handedness have not assessed the reproducibility of their results, and most brain functional studies, except for one fMRI study[19], were done in small samples (<18 left-handers) and thus likely underpowered to provide a reproducible effect of handedness for brain-wide association studies, which require large samples[22].

This cross-sectional study aims to examine the impact of handedness on brain connectivity metrics and its consistency in a sizable group of 9–10-year-old children participating in the ABCD study. We utilized vertex-wise data-driven global functional connectivity density (gFCD) mapping[23] to investigate how handedness influences the extent of functional connectivity hubs throughout the entire brain. Additionally, we conducted a hub-based correlation analysis to evaluate how handedness affects the lateralization of specific functional connectivity pathways. We hypothesized that left-handedness would be associated with higher gFCD in the left-hand area of the motor cortex (right hemisphere) and with lower gFCD in the right-hand area (left hemisphere). Here, we show that left-handedness is associated with increased functional connectivity in the left-hand motor area and reduced connectivity in the right-hand motor area, and with remarkable differences in the laterality of the connectivity of the motor cortex with sensory-motor regions, heteromodal cortical areas, and the cerebellum.

## Results

### Demographics for the ABCD 2.0 data release ($n = 11,875$ children)

The prevalence of handedness for the ABCD cohort corresponded to 79.4% for right-handers, 7.14% for left-handers, and 13.49% for mixed-handers. The prevalence of left-handedness was larger in boys (7.85%) than in girls (6.37%; $\chi^2 = 8.5$, $P = 0.0036$), but mixed-handedness did not differ between boys (13.8%) and girls (13.1%; $\chi^2 = 1.0$, $P = 0.32$). European ancestry was higher for right-handers (0.75) than non-right-handers (0.73; $t = 2.75$, d$f = 3472$, $P = 0.006$). We selected 600 left-handed children and matched them by age, sex, race, scanner, and total brain volume with 600 right-handed and 600 mixed-handed children. There were no differences in head motion and family income as a function of handedness (Table 1).

### Global FCD

To quantify the number of positive edges of each vertex in the dense connectome we computed gFCD with a standard Pearson correlation threshold > 0.6. The average gFCD pattern observed across children in the R and L groups resembled that of adults[24]. Specifically, prominent gFCD hubs were identified in the posterior cingulum, precuneus, occipital, and inferior parietal cortices, consistent with findings from previous research[25]. The remarkably high correlation observed across grayordinates of the gFCD in both the L and R groups ($R > 0.99$) highlights the robust reproducibility of this pattern (Fig. 1).

A vertex-wise t-test revealed higher gFCD, predominantly in the left sensorimotor cortex (Ml), and lower gFCD in the right sensorimotor cortex (Mr) for right- than left-handers (Fig. 2; $P_{FDR} < 0.05$, FDR corrected). Differences in average gFCD within the Ml and Mr ROIs (Fig. 2b) between the R the L groups were highly reproducible in the Discovery and Replication subsamples (Fig. 2c), independently for boys and girls (Fig. S2). Similar analyses of gFCD computed using a correlation threshold $< -0.6$ did not reveal significant group differences in gFCD (Fig. S3), suggesting that the group difference in gFCD is not confounded by negative edges.

After the removal of the effects of sex and race from gFCD using grand mean scaling, a normalized difference between average gFCD values in Ml and Mr provided a neurobiological-based index of handedness,

$$\text{Handedness index} = \frac{\text{gFCD(Ml)} - \text{gFCD(Mr)}}{\text{gFCD(Ml)} + \text{gFCD(Mr)}} \qquad (1)$$

which differentiated left-handers from right-handers (Cohen's $d = 0.75$) and mixed-handers (Cohen's $d = 0.53$; Fig. 2e) and was significantly correlated with the children's handedness scores (Edinburgh Handedness Inventory Short Form) both in the discovery ($r = 0.37$) and replications ($r = 0.26$) cohorts (Fig. 2f). Note that these large effect sizes contrast with the medium effect size of the contrast on gFCD between left-handers and right-handers (Cohen's $d = 0.35$).

We used the activation patterns to the left- and right-hand movements, averaged across 997 healthy adults[26], to assess the homology of Ml and Mr regions in the right and left hemispheres. Using a stringent Cohen's $d > 1.3$ thresholds, we found that the brain activation patterns for right- and left-hand movements distinctly encompassed the Ml and Mr ROIs, respectively (Fig. 3a, b). This highlights the specificity of the group differences in gFCD related to the left-hand (Mr) and right-hand (Ml) regions of the motor cortex.

### Ipsilateral and contralateral gFCD

To rule out the effects of homotopic functional connectivity, we computed ipsilateral (intra-hemisphere) and contralateral (inter-hemisphere) gFCD components by restricting the calculation to the same (ipsilateral) or the opposite (contralateral) hemisphere of each vertex. While the patterns of intra-hemispheric gFCD (Fig. 3c) were like those in Fig. 1, the patterns of inter-hemispheric gFCD highlight strong hubs of contralateral connectivity in the inferior motor and occipital cortices, the occipitoparietal junction, rectal gyrus, precuneus, and the posterior cingulum (Figs. 3d and S4). The intra-hemispheric gFCD was lower in Ml and higher in Mr for L than R ($P_{FDR} < 0.05$), such that the Handedness index differentiated the R, L, and M subgroups (Fig. 3c). There was no significant difference in inter-hemispheric gFCD between L and R in any brain region, and the Handedness index did not differentiate the subgroups (Fig. 3d) suggesting that homotopic connectivity did not drive the group differences in gFCD in the hand-motor cortex.

### Connectivity of the hand motor area

We used gFCD-guided seed-voxel correlation analyses to explore the functional connectivity patterns of the gFCD clusters (Ml and Mr). In right-handers, the Ml and Mr seeds in Brodmann area 3 had strong average rsFC in the left and right premotor and sensorimotor cortices, respectively (Fig. S5). The seeds also had significant though weaker average connectivity in contralateral premotor and somatomotor areas, opercular area 4, retro insular cortex, and ipsilateral regions of the anterior (lobe V) and posterior (lobe VIII) cerebellum (CER; Fig. S6), consistent with prior studies[27].

**Table 1 | Demographics of *Discovery* and *Replication* subsamples of right-handers (R), left-handers (L), and mixed-handers (M)**

| | R | | L | | M | | P |
|---|---|---|---|---|---|---|---|
| | Discovery | Replication | Discovery | Replication | Discovery | Replication | |
| Girls | 130 | 127 | 130 | 127 | 130 | 127 | n.s.[†] |
| Boys | 173 | 170 | 173 | 170 | 173 | 170 | |
| Mean age (SD) [years] | 10.00(.62) | 9.95(.63) | 10.00(.62) | 9.95(.63) | 9.99(.62) | 9.93(.64) | n.s.[*] |
| African American | 55 | 34 | 55 | 34 | 55 | 36 | n.s.[†] |
| Asian | 4 | 8 | 4 | 8 | 2 | 4 | |
| Hispanic | 55 | 52 | 55 | 52 | 56 | 53 | |
| White | 161 | 174 | 161 | 174 | 162 | 174 | |
| Other | 28 | 29 | 28 | 29 | 28 | 30 | |
| Mean brain volume (SD) [mL] | 1211(105) | 1221(114) | 1211(106) | 1220(120) | 1214(106) | 1224 (111) | n.s.[*] |
| Family income | 7.4 (2.4) | 7.6 (2.2) | 7.1 (2.5) | 7.4 (2.6) | 7.1 (2.4) | 7.3 (2.4) | n.s.[*] |
| Mean FD(SD) [mm] | 0.12 (0.04) | 0.11 (0.04) | 0.12 (0.04) | 0.12 (0.04) | 0.12 (0.04) | 0.12 (0.04) | n.s.[*] |
| Siemens | 192 | 200 | 192 | 200 | 193 | 200 | n.s.[†] |
| GE | 75 | 58 | 75 | 58 | 74 | 58 | |
| Phillips | 36 | 39 | 36 | 39 | 36 | 39 | |

*p*: 2-sided statistics using $\chi^2$-test[†], analysis of variance[¥] or covariance[*].

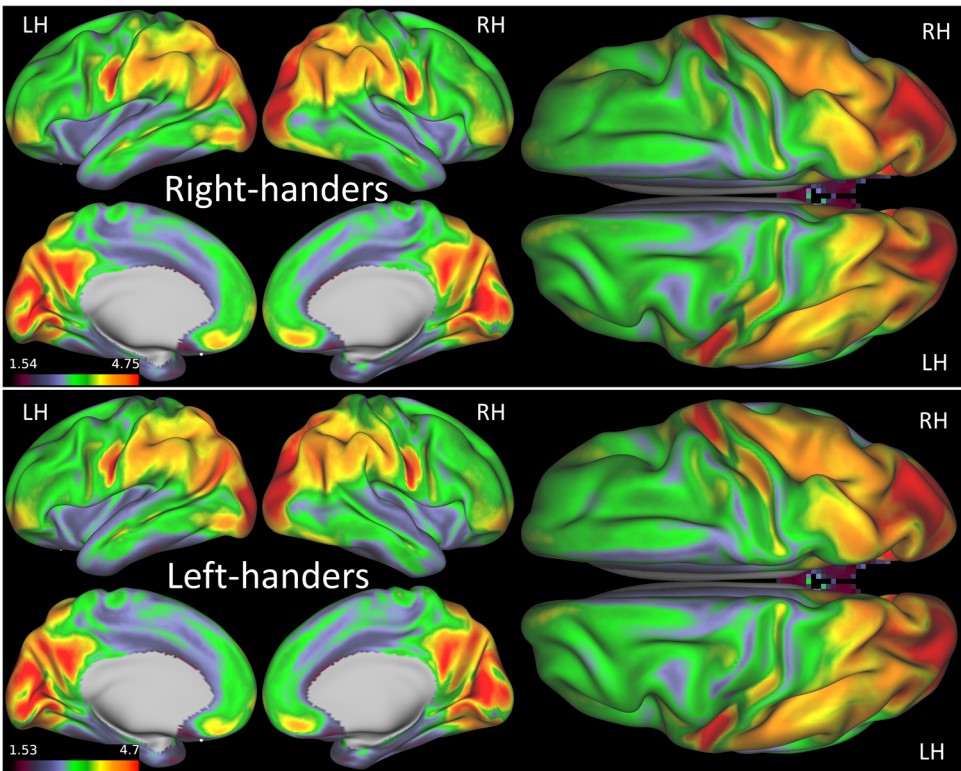

**Fig. 1 | gFCD patterns.** Average global functional connectivity density (gFCD) maps overlaid on inflated lateral, medial, and dorsal surfaces of the left (LH) and right (RH) cerebral hemispheres across 600 right- and 600 left-handed children. Source data are provided as a Source Data file.

## Functional lateralization

Like the handedness index, we used a normalized laterality index:

$$\text{Laterality index} = \Delta = \frac{rsFC(Ml) - rsFC(Mr)}{|rsFC(Ml)| + |rsFC(Mr)|}, \quad (2)$$

to assess differential connectivity strength with the hand-motor areas while accounting for potential variations in the overall strength of connectivity in the whole brain. Overall, the connectivity of the hand-motor area had a similar laterality pattern between right and left-handers (Fig. 4). Specifically, somatomotor, premotor, mid-cingulate, opercular, and retro insular cortices exhibited a pronounced positive Δ in the left cortical hemisphere and a negative Δ in the right hemisphere. In CER, the Δ was positive in the right hemisphere and negative in the left hemisphere.

## Effect of handedness on lateralization

We used vertex-wise t-test analysis to identify brain regions showing between-group differences in the laterality index. This analysis showed that the degree of lateralization differed between left-

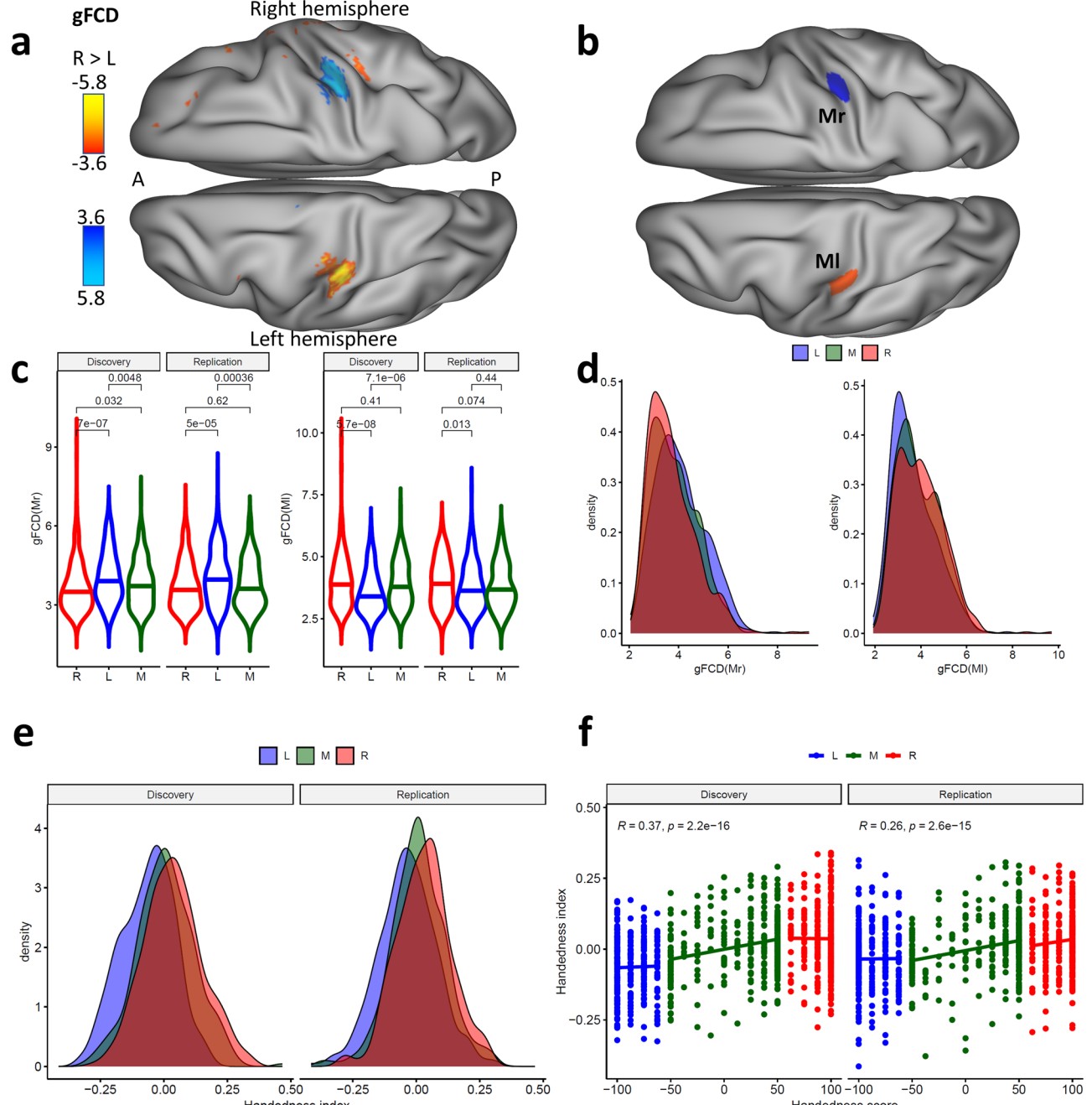

**Fig. 2 | Handedness and global functional connectivity density (gFCD).** Statistical differences (t-score) in resting-state gFCD between right (R) and left-handers (L) (**a**) or the left (Ml) and right (Mr) somatomotor ROIs (**b**) in Brodmann area 3, superimposed on inflated dorsal views of a human brain template. Violin plots show that compared to right- and mixed-handers (M), left-handers had lower average gFCD in the Ml (right panel) and higher average gFCD in Mr (left panel), both in the Discovery ($n = 909$) and Replication ($n = 891$) samples (**c**). Density plots show the distributions of average gFCD in the Ml (right) and Mr (left) ROIs (**d**) and the handedness index (**e**) for L ($n = 600$), R ($n = 600$), and mixed handers (M, $n = 600$). **f** Reproducibility of the linear association between the handedness index and the children's handedness scores across L, R, and M in the Discovery and Replication samples. Statistical model: two-sample $t$-test (2-sided). An FDR-corrected threshold $P_{FDR} < 0.05$ (2-sided) was used to display the statistical maps. Source data are provided as a Source Data file.

handers and right-handers (Fig. 5). Specifically, left-handers had lower Δ than right-handers, bilaterally, in areas 6d (premotor), BA 1 (primary somatosensory), 24dd (mid cingulate), and POS2 (parieto-occipital sulcus, area 2), the cerebellar lobules V and VIII (CER), and other brain regions (Fig. 5; $P_{FDR} < 0.05$). Conversely, Δ was higher for left- than right-handers, bilaterally, in FST (fundus of the superior temporal visual area), BAs 4 (primary motor) and 40 (inferior parietal), IPS1 (intraparietal sulcus, area 1), and other brain regions (Fig. 5; $P_{FDR} < 0.05$).

The differential connectivity with the hand motor areas (Ml and Mr) varied across groups and reproduced in the Discovery and Replication subsamples in all ROIs (Fig. 6a, b, and Figs. S7–S10) and was a valuable tool for the interpretation of the group differences in connectivity. Specifically, the ROI analysis showed that right-handers and left-handers had opposite Δ bilaterally in FST, IPS1, and BA 40, and in CER and area 6d of the left hemisphere and BA4 in the right hemisphere (Fig. 6c, d). In all ROIs, mixed-handers had intermediate Δ values between those of right- and left-handers, such that handedness

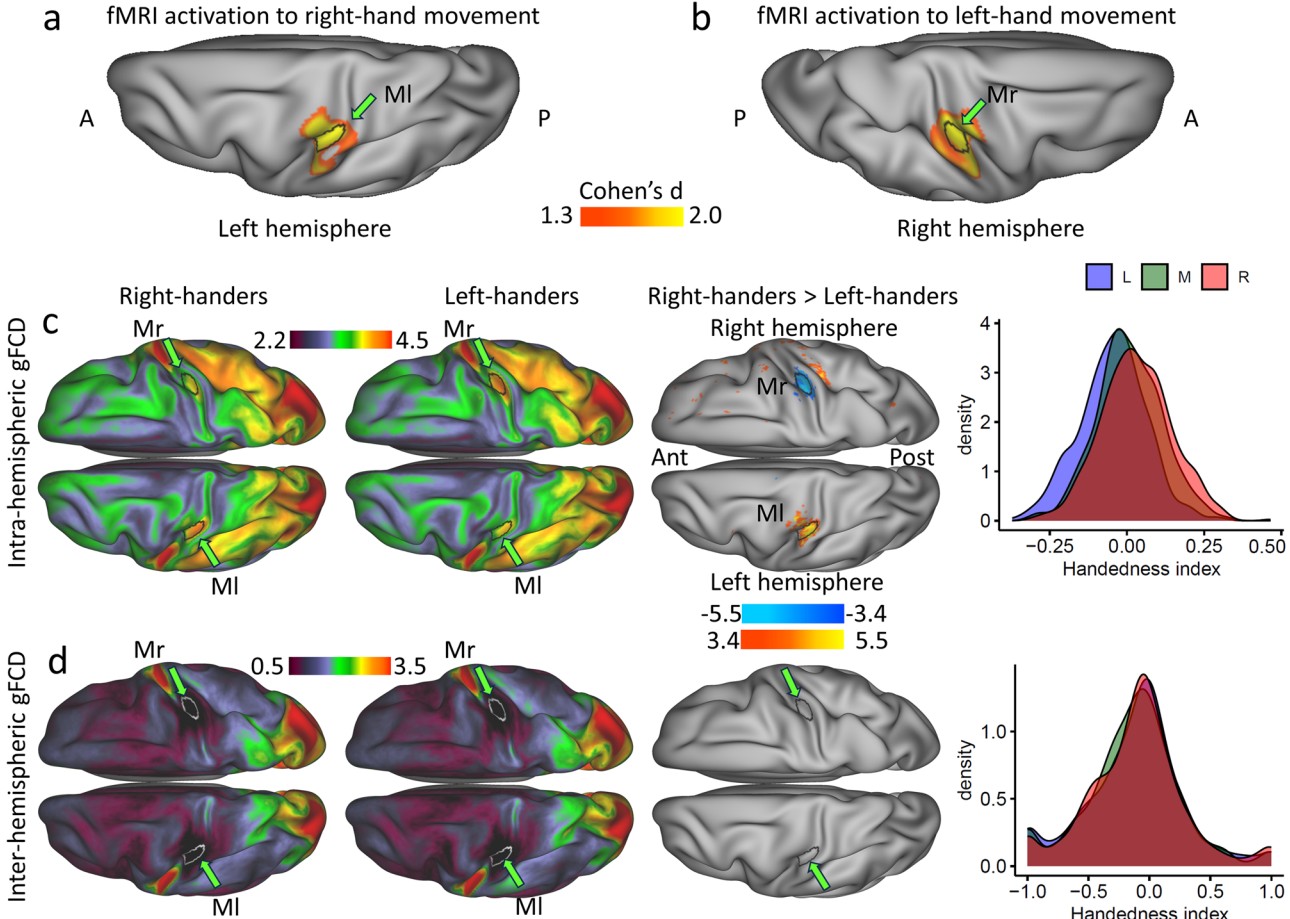

**Fig. 3 | Brain activation to hand movements and ipsilateral and contralateral gFCD.** Overlap between left (Ml) and right (Mr) sensorimotor ROIs and brain activation patterns to the right- and left-hand movements, respectively (**a** and **b**). Intra (**c**) and inter (**d**) hemispheric global functional connectivity densities (gFCD) mapped on inflated dorsal views of the cerebral surface for 600 right-handers and 600 left-handers, and their statistical differences (t-score). The density

plots show the corresponding distributions of handedness index (Eq. 1) computed from measures in the left-hand (Mr) and right-hand (Ml) motor ROIs for L, R, and 600 mixed-handers (M). Statistical model: Two-sample T-test (2-sided). An FDR-corrected threshold $P_{FDR} < 0.05$ (2-sided) was used to display the statistical maps. Green arrows and black or white line contours highlight the location of the Ml and Mr ROIs from Fig. 2. Source data are provided as a Source Data file.

was associated with pronounced Δ-decreases in 6d, CER, 24dd, and BA 1, and with Δ-increases in FST, IPS1, and BAs 4 and 40, both in the left and right hemispheres (Fig. 6c, d). The effect of handedness on Δ was highly reproducible across the Discovery and Replication subsamples (Fig. 6c, d).

## Specialization

To characterize differences in Δ between right- and left-handers in terms of the functional processing hierarchy, we computed a functional specialization index based on multi-modal parcellation of the human cerebral cortex[26]. The functional specialization index differentiated unimodal cortical areas (e.g., visual, auditory, and sensorimotor cortices) with a high functional specialization index (>0.5) from heteromodal association cortical areas (e.g., insula and dorsolateral prefrontal and inferior parietal cortices) with lower specialization index (Figs. 5 and S1). Most of the clusters showing significant differences in Δ between right- and left-handers were in heteromodal association cortices. Specifically, 85% of the grayordinates with significant differences in laterality between right-handers and left-handers, including most major clusters (areas 40 and 24dd, POS2, FST, IPS1), were in regions with functional specialization index < 0.5 (Fig. 5).

## gFCD asymmetry

We studied brain functional connectivity asymmetry by subtracting the gFCD values in the left cortical hemisphere (LH) from the

corresponding grayordinates in the right hemisphere (RH), gFCD asymmetry = gFCD(RH) − gFCD(LH), for each individual, such that positive asymmetry reflected rightward asymmetry while negative asymmetry reflected ones leftward asymmetry.

In right-handed individuals, we observed that the left hemisphere showed predominantly higher gFCD in regions within the somatosensory and motor cortices, lateral middle and inferior temporal cortex, anterior cingulate, and medial prefrontal cortices, while it exhibited lower gFCD in regions within the insula, premotor cortex, paracentral lobular cortex, and midcingulate cortex ($P_{FDR} < 0.05$; Fig. 7a). Remarkably, we found a similar and consistent pattern in right-handers, left-handers, and individuals with mixed hand preference (Dice coefficient > 0.71; Fig. S11). Notably, in right-handers, gFCD did not differ between the left and right hemispheres in the hand-motor area (Fig. 7a). Conversely, left-handers exhibited a pronounced rightward asymmetry in the hand-motor area, displaying significantly higher gFCD in the right hemisphere compared to the left (Fig. 7b). These differences were highly significant, encompassing the entire Mr ROI (Fig. 7c).

## Bain structure

There were no significant effects of handedness on brain morphometrics (sulcal depth, cortical thickness, and curvature) or cortical myelin (Fig S12), as well as on white matter diffusion metrics (fractional anisotropy, FA, mean, MD, longitudinal, lD, and transverse, tD, diffusivities).

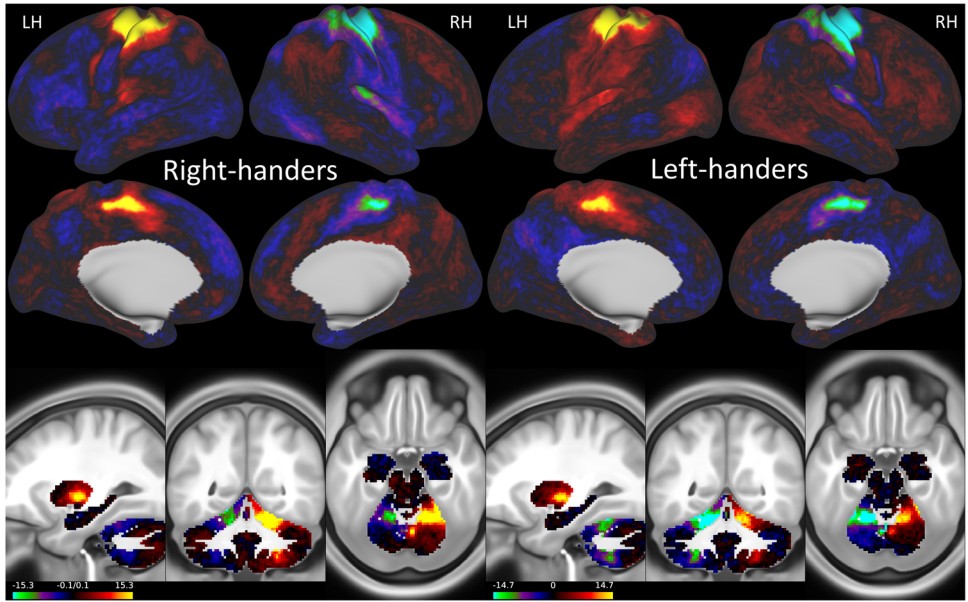

**Fig. 4 | Laterality index.** Statistical significance (t-score) of the laterality index, Δ, for 600 right-handers and 600 left-handers, superimposed on inflated lateral and medial surfaces of the left (LH) and right (RH) cerebral hemispheres and 3 orthogonal brain views showing the differential connectivity patterns in subcortical regions and cerebellum. Source data are provided as a Source Data file.

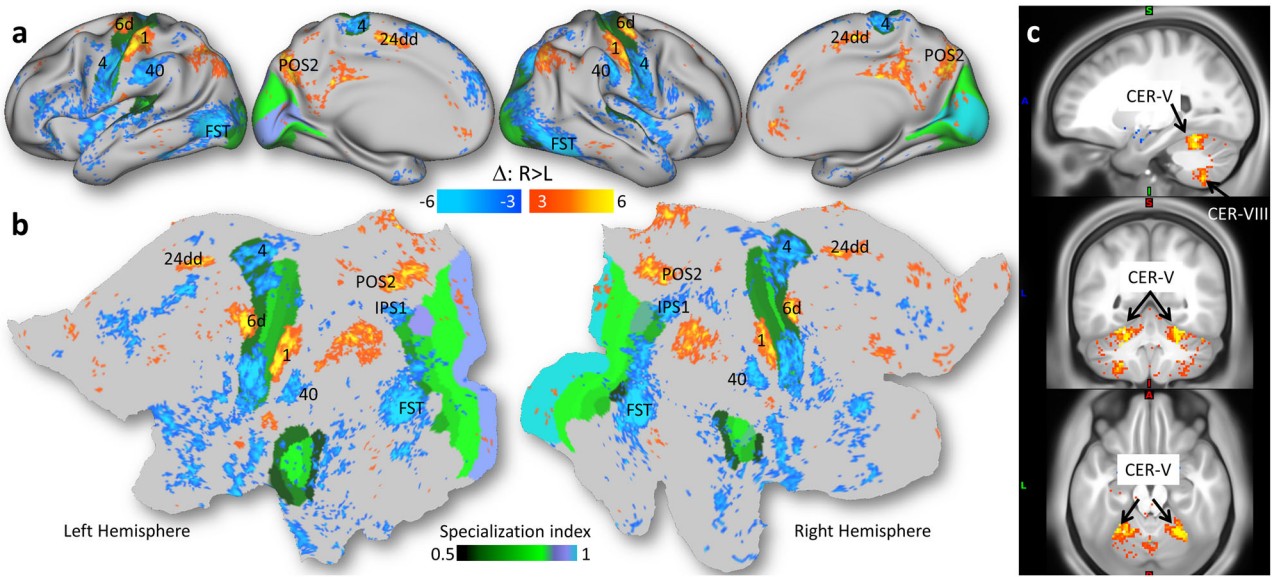

**Fig. 5 | Laterality index and handedness.** Statistics (t-score) for the difference in the laterality index (Δ; Eq. 2) between right-handers (R, n = 600) and left-handers (L, n = 600) and the score of a functional specialization index (see text) superimposed on inflated (**a**) and flat (**b**) views of a human brain template and 3 orthogonal MRI views of the human brain (**c**). Statistical model: Two-sample T-test (2-sided). An FDR-corrected threshold $P_{FDR} < 0.05$ (2-sided) was used to display the statistical maps. 6d: premotor area d; FST: fundus of the superior temporal visual area; CER: cerebellum lobes V and VIII; 24dd: mid cingulum; POS2: parieto-occipital sulcus, area 2; IPS1: intraparietal sulcus area 1; Brodmann areas 1: somatosensory, 4: superior motor, 6d: premotor, and 40: inferior parietal. Source data are provided as a Source Data file.

We also investigated the effect of handedness on brain asymmetry by comparing MRI metrics (sulcal depth, curvature, cortical myelin, and thickness) and white matter diffusion metrics (fractional anisotropy, FA, mean, MD, longitudinal, lD, and transverse, tD, diffusivities) between the right and left hemispheres, as for gFCD asymmetry. Much like the asymmetry patterns observed for gFCD, the asymmetries of these metrics were highly consistent across right-handers, left-handers, and mixed-handers with Dice coefficients > 0.86 (curvature), .65 (myelin), 0.93 (sulcal depth), and 0.78 (cortical thickness) (Figs. S13–S16). Similarly, the asymmetries of DTI metrics were highly

consistent across right-handers, left-handers, and mixed-handers (Fig. S17). However, the patterns of structural asymmetry did not reveal statistically significant effects of handedness (Fig. S18). Similarly, the asymmetry of the white matter diffusion metrics did not reveal significant effects of handedness (Fig. S19).

## Discussion
A better understanding of the lateralized organization of the human brain during development can be gained by examining the relationship between handedness and brain functional and structural connectivity

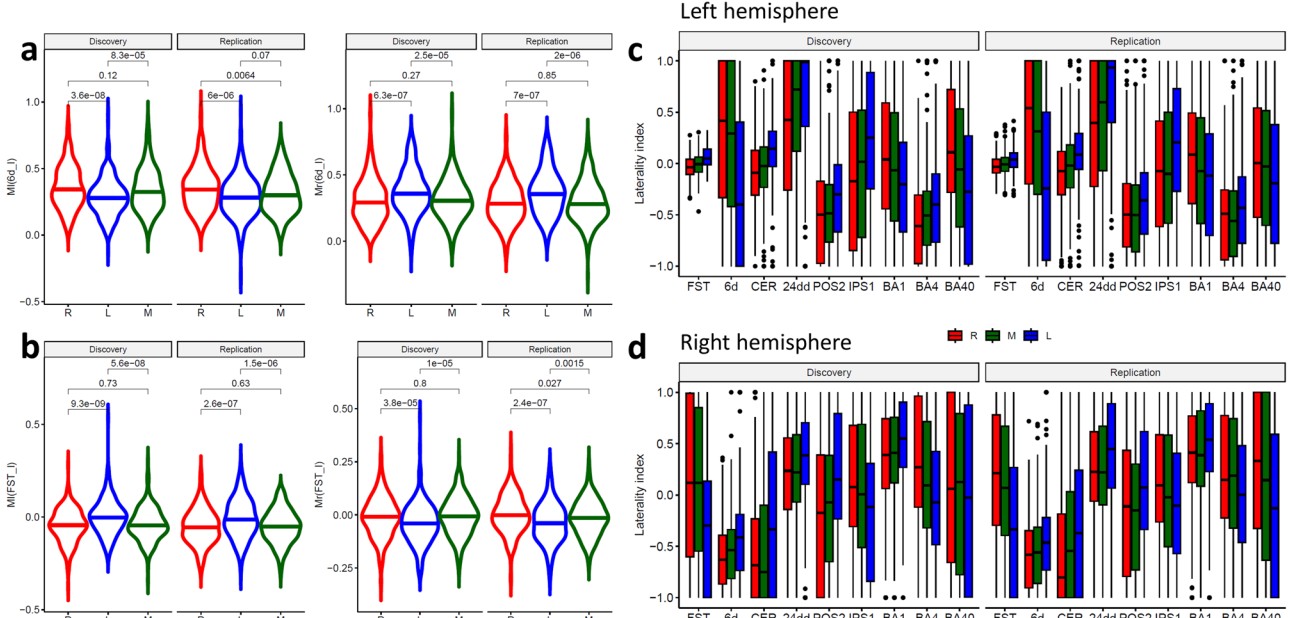

**Fig. 6 | ROI analyses.** Violin plots showing that the average resting-state functional connectivity (rsFC) of the right-hand (Ml) motor seed in the left premotor ROI (6d_l; **a**) was lower and that of the left-hand (Mr) motor seed higher for left- (L, n = 600) compared to right- (R; n = 600) and mixed-handers (M; n = 600). Conversely, the average rsFC of the Ml seed in the fundus of the superior temporal visual area (FST; **b**) ROI was higher and that of the Mr seed lower for L than for R or M. Findings reproduced in the Discovery (n = 909) and Replication (n = 891) samples. Distribution of the laterality index (Eq. 2) for different groups and ROIs in the left (**c**) and right (**d**) hemispheres. The box extends from the lower to the upper quartile (25th to 75th percentile) of the data, with the horizontal line representing the median. The whiskers extend to 1.5 times the interquartile range from the lower and upper quartiles, and outliers beyond this range are represented as individual points. Statistical model: Two-sample T-test (2-sided). CER: cerebellum lobe V; 24dd: mid cingulum; POS2: parieto-occipital sulcus, area 2; IPS1: intraparietal sulcus area 1; Brodmann areas 1: somatosensory, 4: superior motor, 6d: premotor, and 40: inferior parietal. Error bars are standard errors of the means. Source data are provided as a Source Data file.

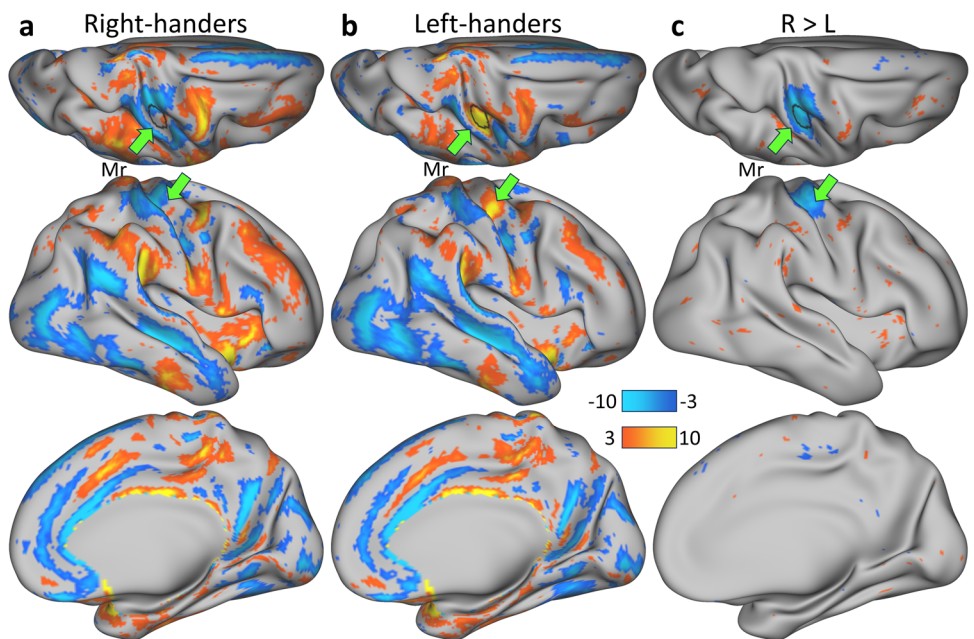

**Fig. 7 | gFCD asymmetry.** Statistical differences (t-scores) in global functional connectivity density (gFCD) between corresponding gray ordinates in the right and left cortical hemispheres for 600 right-handed (R; **a**) and 600 left-handed (L; **b**) children, as well as the statistical group differences in gFCD asymmetry (**c**) superimposed on superior, lateral, and medial views of the right cerebral hemisphere. The green arrow points to the region of interest in the hand motor cortex (Mr; black contour). Statistical model: One-sample (a and b) or two-sample (c) T-test (2-sided). An FDR-corrected threshold $P_{FDR} < 0.05$ (2-sided) was used to display the statistical maps. Source data are provided as a Source Data file.

during childhood. In this study, we present findings from a large group of 9- to 10-year-old boys and girls in whom we compared the functional connectivity in hand-motor cortices. We demonstrate that left-handedness is associated with increased functional connectivity in the left-hand motor area (Mr) and reduced connectivity in the right-hand motor area (Ml). Additionally, we observed differences in the laterality of connectivity with sensory-motor regions, heteromodal cortical areas, and the cerebellum. This handedness-related effect on

connectivity laterality was consistent in both the Discovery and Replication subgroups. While the hand-motor network is similar in left-handed and right-handed children, there are notable differences in the asymmetry of their connectivity patterns.

In a previous study utilizing data-driven gFCD, we showed the lateralization of Broca's language area in healthy adults[28]. Here we used gFCD to pinpoint the location of brain connectivity hubs that had differential connectivity in 600 left-handed compared to 600 right-handed children. This revealed stronger gFCD strength in the left-hander's middle portion of the right somatomotor cortex, consistent with the fMRI activation caused by left-hand movements[19,27], and weaker gFCD in their left motor cortex area (middle portion), also consistent with fMRI activation caused by right-hand movements[19,27]. These findings are also consistent with the differences in cerebral blood flow changes induced by handwriting between right-handers and converted left-handers[29]. This stronger representation of the left hand in the lefthander's brain than in the righthander's brain and vice versa also corresponds with the lateralization of language[30], which has a leftward specialization for comprehension and speech in 97% of right-handers but only in 70% of left-handers[1]. The handedness index, which was defined as the relative difference in gFCD between the right- and left-hand motor areas similar to the previously proposed language asymmetry index[31], provided a neurobiological-based index of handedness (IH; Eq. (1)) with increased effect size (Cohen's d = 0.75) that was significantly associated with the children's handedness scores (Edinburgh Handedness Inventory Short Form). The IH serves as a neuroimaging equivalent to Oldfield's laterality quotient[32], enhancing the sensitivity of neuroimaging studies on handedness and enabling investigations in smaller sample sizes.

The overlaps between the effect of handedness on resting-state gFCD in ABCD children and the fMRI activation pattern to the right- and left-hand movements in young adults from the HCP[26] in Ml and Mr link the observed group differences in gFCD to specific motor regions associated with handedness, providing a more detailed understanding of the neural basis for motor function. While the group differences in gFCD were consistent with those in intra-hemispheric gFCD, there were no differences in inter-hemispheric gFCD between left- and right-handed children. This implies that in left-handed children, the left-hand motor region (Mr) had higher connectivity with regions of the right hemisphere than in right-handed children. This discovery implies that hand-motor function relies predominantly on intra-hemispheric connectivity within the left hemisphere for right-handers, as previously noted[33], but within the right hemisphere for left-handers.

Multiple brain areas demonstrated pronounced lower lateralization of hand-motor connectivity (Δ) for left- than right-handers, including unimodal sensorimotor areas (6d and BA 1) and motor area 24dd in the midcingulate cortex, that replicated in Discovery and Replication subsamples. The left-handers' lower lateralization in the primary somatosensory cortex (BA 1) and their higher lateralization in the superior and inferior parts of BA 4 [foot and tong areas of the motor cortex[27]] compared to that of right-handers is consistent with the association between interhemispheric asymmetries and handedness such that in right-handers the left central sulcus is deeper than the right, and vice versa in left-handers[34]. The left-handers' lower rightward lateralization (Δ > 0) in the superior premotor area 6d, which was shown to be activated by hand movements[26], is consistent with the difference in motor-premotor effective connectivity between right- and left-handers reported by fMRI studies using a hand motor task[20], and with the lower surface area asymmetry in the precentral and midcingulate cortices for 3062 left-handers than 28,802 right-handers[14].

Heteromodal brain areas (24dd, POS2, FST, BA40, and IPS1) also demonstrated differential lateralization with the hand-motor areas (Δ; Eq. 2) for right- and left-handers suggesting that handedness impacts regions beyond the motor pathways. Compared to right-handers, left-handers demonstrated lower rightward lateralization (Δ > 0) in the midcingulate motor cortex area 24dd, which is involved in upper limb movements and is consistent with findings of lower surface area asymmetry in the left-handers' mid cingulum[14], and in the parieto-occipital sulcus area POS2 that was shown to be activated by motor cues[26]. Conversely, left-handers had higher leftward lateralization (Δ < 0) than right-handers in the visual area FST and the dorsal stream area IPS1, which were not engaged by motor tasks and showed activation to a social task[26], and in the inferior parietal cortex area BA40 that is involved in attention[35]. Overall, the pattern of group differences in Δ was similar to the pattern of differences in fMRI signals between right- and left-handers during a motor task[17,18].

Here we also document reproducible group differences in cerebellar asymmetry, such that left-handers had a weaker rightward lateralization index (Δ, Eq. 2) in the left cerebellar hemisphere and stronger leftward lateralization in the right hemisphere than right-handers. The connectivity patterns of the hand-motor areas in the cerebellum are consistent with those reported by prior studies on contralateral cortico-cerebellar functional connectivity[19,27,36]. However, our findings of lower rightward lateralization of hand-motor connectivity in the cerebellum contradict the lack of cerebellar functional lateralization and connectivity with respect to hand-motor control[37].

The asymmetries of the structural brain metrics (cortical myelin, thickness, curvature, sulcal depth, FA, MD, lD, and tD) were consistent across children with varying hand preferences, and there were no differences in laterality between left- and right-handed children. This contrasts with the differences in asymmetry of gFCD in the hand motor areas, indicating that functional connectivity better reflects hand motor behaviors than structural measures.

Prior fMRI studies have shown increased brain lateralization as children age[9,38]. In this study, we could not assess the effect of age on brain asymmetry due to the narrow age range in the baseline ABCD sample. However, the longitudinal nature of the ABCD study will enable future studies to investigate age-by-handedness interactions on Δ and to assess if laterality differences in structural measures emerge as these children transition into adulthood. Neuroimaging studies using UK biobank data have reported reduced leftward asymmetry in cortical thickness asymmetry in the postcentral gyrus for 3062 left-handed adults than for 28,802 right-handed adults[14], which highlights a small effect of handedness on cortical thickness (Cohen's d = 0.04) and perhaps other structural metrics. The moderate-large effect size of handedness on gFCD and the lack of significant associations with any of the morphometrics or white matter diffusion metrics likely reflect the relatively small sample in the present study. The lack of significant differences in diffusion metrics between left- and right-handed children in this study is consistent with the absence of differences in white matter microstructure between right- (n = 2646) and left-handed (n = 293) children[16] but does not support the lower anisotropy for left-handed adults (n = 40)[15]. Similarly, the lack of effects of handedness on cortical thickness contrasts with the lower rightward asymmetries reported for ten non-dextral healthy adults[10]. Some of these discrepancies could reflect the effects of brain development and/or differences in statistical power between studies. For instance, in our investigation, the percentage of left-handed children (7.14%) was lower, while the proportion of mixed-handed individuals (13.49%) was higher compared to figures reported in studies involving adults (9.33% for left-handedness and 10.6% for mixed-handedness)[3]. This suggests that at baseline, ABCD children may not have fully established their handedness, and some may still be in the process of developing a dominant hand.

Here we show reproducible effects of handedness on resting-state functional connectivity but not structural connectivity metrics or morphometrics in 9- to 10-year-old children and provide a simple neurobiological index of handedness that shows a strong correlation with handedness scores in children. The alterations in

interhemispheric motor connectivity balance highlight distinct neural organizations in left-handers and right-handers. Furthermore, the contrasting connectivity patterns observed between the hand-motor area and the visual and primary motor areas underscore the intricate interplay among brain regions in individuals with different handedness. Our findings also demonstrate that functional connectivity differences in the hand-motor cortex are already strongly expressed in childhood.

## Methods

### Participants

The adolescent brain cognitive development (ABCD) study is a longitudinal investigation conducted across multiple sites, involving over 11,800 children aged 9–10 years with demographic characteristics representative of the broader US population[39]. Children included in the study were proficient in English and were excluded if they had any medical, neurological, or cognitive conditions, exhibited poor English-language skills, or had contraindications for undergoing MRI scans[40]. Written informed consent was obtained from parents, with children also providing written assent for their involvement. The ABCD Study received approval from the institutional review board (IRB) at the University of California in San Diego and obtained local IRB approval at 21 data collection sites nationwide[41].

Utilizing scores from the Edinburgh Handedness Inventory Short (EHIS) form, we identified a cohort of 600 left-handed children (L) from the ABCD 2.0 data release[42], for whom resting-state fMRI data in Connectivity Informatics Technology Initiative (CIFTI) format were available (Table 1). To ensure comparability, we employed the matchControls function of the e1071 R package to pair the left-handers with 600 right-handed (R) and 600 mixed-handed (M) children from the ABCD 2.0 data release who also had resting-state fMRI data in CIFTI format. Participants displaying excessive head motion during resting-state fMRI scans (>50% of time points with framewise displacement (FD) < 0.5 mm) were excluded from the L, R, and M groups. Given the variability of diffusion metrics across MRI scanners in the ABCD study[43], our analysis of structural connectivity metrics was confined to a subset of 1177 participants (392 left-handed, 392 right-handed, and 393 mixed-handed) who underwent MRI scans on Siemens scanners. This excluded 623 participants who underwent MRI scans on GE (N = 398) or Phillips (N = 225) scanners.

### Handedness

The handedness score rating (1: righthanded; 2: lefthanded; 3: mixed-handed) documented in the ABCD Youth EHIS form was used for group classification. We computed the average handedness score as the average frequency of hand use for writing, throwing, and spoon and toothbrush use (always right: 100; usually right: 50; both equally: 0; usually left: −50; always left: −100) documented in the EHIS form, which was downloaded from the National Institute Mental Health Data Archive (NDA; https://nda.nih.gov/).

### MRI data

3 T MRI scanners (Siemens Prisma, Phillips, and General Electric 750) equipped with adult-size multi-channel coils capable of multiband echo planar imaging (EPI) and harmonized data acquisition protocols were used for MRI[43,44]. 3D T1w inversion prepared RF-spoiled gradient echo and T2w variable flip angle fast spin echo pulse sequences with 1 mm isotropic resolution were used for structural MRI. Multiband echo-planar imaging (EPI)[45,46] with slice acceleration factor = 3, five *b*-values ($b = 0$, 500, 1000, 2000, and 3000 s/mm$^2$), 96 diffusion directions, and 1.7 mm isotropic resolution was used for diffusion MRI acquisition[43]. T2*-weighted multiband EPI (TE/TR = 30/800 ms, 2.4 mm isotropic resolution, 60 slices covering the entire brain, slice acceleration = 6, and flip angle = 52°) was used to acquire functional MRI (fMRI) data with blood-oxygen-level-dependent contrast[44].

Publicly available datasets from the ABCD brain imaging data structure (BIDS) Community Collection (ABCC) (https://collection3165.readthedocs.io/en/stable/), which include resting-state fMRI data in CIFTI format[47] from 10,038 children that have passed quality assurance, were used for the analyses of functional connectivity, myelin GIFTI shape maps computed from the ratio of T1w/T2w image intensities in the voxels between the white and pial surfaces and mapped onto the mid thickness surface without spatial smoothing[48], and brain morphometrics (cortical curvature, thickness, and sulcal depth) derived from MRI T1-weighted scans with FreeSurfer and converted to GIFTI shape files[49]. The ABCD-BIDS pipelines minimized unwanted variability from differences in MRI scanners (GE, Phillips, and Siemens). For structural connectivity analyses, we downloaded from NDA and used tabulated diffusion imaging metrics (fractional anisotropy, FA, mean, MD, longitudinal, lD, and transverse, tD, diffusivities)[43].

### Reproducibility

Participants were divided into two separate demographically matched subgroups: the Discovery sample (N = 909, comprising 303 left-handed, 303 right-handed, and 303 mixed-handed individuals) and the Replication sample (N = 891, consisting of 297 left-handed, 297 right-handed, and 297 mixed-handed individuals). This division was achieved using the ABCC's "matched group" designation, which considers sociodemographic factors known to influence brain development, such as age, sex, ethnicity, grade level, the highest level of parental education, and handedness[49]. Importantly, there were no notable discrepancies in brain volume, age, framewise displacement (FD), or the distribution of MRI manufacturers and racial/ethnic groups between the Discovery and Replication subgroups across the left-handed (L), right-handed (R) and mixed-handed (M) categories (Table 1).

### MRI data processing

The ABCD-BIDS pipeline[49] is like the HCP pipeline. However, it performs the nonlinear registration to the standard atlas using ANTS[50], which consistently outperforms other nonlinear registration methods[51]. Furthermore, the fMRI pre-processing steps in the ABCD-BIDS pipeline perform standard denoising by regressing out time-varying head motion, white matter, and CSF signals, and the global signals that may impact group comparisons[52,53] and separates fictitious motion induced by breathing-related magnetic field changes from true head motion[54]. Specifically, 5 consecutive pipelines perform brain extraction, denoising, and normalization of structural data to a standard template (PreFreesurfer); brain segmentation and creation of cerebral surfaces (Freesurfer); conversion of structural data to CIFTI format (PostFreesurfer); registration of the functional time series to the volumetric standard template (fMRIVolume); and conversion of functional time series data to the CIFTI format (fMRI-Surface). A probabilistic approach for automated segmentation of white matter fiber tracts[55] while excluding gray matter (GM) and cerebral spinal fluid (CSF) voxels was used to quantify diffusion MRI metrics (fractional anisotropy, FA, and mean, MD, longitudinal, lD, and transverse, tD, diffusivities)[43]. Specifically, diffusion MRI pre-processing included eddy current correction along the phase-encode direction[56–59] rigid-body-registration, minimization of spatial and intensity distortions[43] registration of b0 images to T1w images, and 1.7 mm resampling.

### Quality assurance (QA)

The ABCD study used automated QA procedures[43]. Specifically, images were corrected for scanner-specific gradient distortions and intensity inhomogeneity. Trained raters inspected images for poor quality and artifacts such as blurring, ghosting, or ringing that could prevent brain segmentation.

## Head motion

Time frames with excessive head motion (FD > 0.5 mm) were removed using motion censoring information estimated with the ABCD-BIDS pipeline. Because head motion is a serious concern for pediatric structural and functional neuroimaging studies[60–62], the statistical analysis further controlled for the subjects' tendency to move their head while resting in the scanner (see below).

## gFCD

The Pearson correlation was used to map the gFCD strength at a given grayordinate, $x_0(t)$, from 0.01 to 0.10 Hz band-pass filtered CIFTI time series with $N = 91,282$ grayordinates[47] and a maximum of 1520 time points (20 min). Specifically, the gFCD was computed as the logarithm of the total number of edges between $x_0$ and all other 91,281 grayordinates in the brain[63] using a correlation $R > 0.6$[23,64]. This calculation was repeated for all $x_0$ grayordinates in the brain[65] using Matlab version R2023a (MathWorks, Inc., Natick, MA) and the Beowulf cluster at NIH.

## rsFC

The connectome workbench function cifti-average-roi-correlation was used to map the resting-state functional connectivity (rsFC) of the right-hand (Ml) and left-hand (Mr) seeds, which demonstrated the strongest effects of handedness on gFCD, from motion-corrected and 0.01–0.10 Hz band-pass filtered CIFTI time series. Specifically, the time-varying fMRI signals were averaged across gray ordinates within Ml (center vertex# 7956, 10-mm radius, left hemisphere) and Mr (center vertex# 7985, 10-mm radius, right hemisphere) motor cortex seeds independently. The Pearson correlation and the Fisher transformation were used to compute normalized correlation maps independently for Ml and Mr.

## Regions of interest (ROIs)

Brain clusters showing peak group differences in the laterality of the resting-state functional connectivity of the hand-motor area, Δ, were selected to extract average rsFC values within 10-mm radius ROIs for each participant (Table S1). The multi-modal parcellation of the human cerebral cortex with 360 cortical and 19 subcortical partitions, which reflected differences in brain function, connectivity, and/or topography[26], was used to interpret the effect of handedness on rsFC.

## Functional specialization index

To determine the overall functional specialization of the ROIs, we used the RGB color scheme in the multi-modal parcellation of the human cerebral cortex, which informs the ROIs' degree of association with 3 functionally specialized domains: auditory (red), sensorimotor (green), and visual (blue)[26]. Specifically, the functional specialization index was defined in terms of the absolute differences in specialization between domains $S_1$ = auditory vs. somatosensory; $S_2$ = auditory vs. visual; and $S_3$ = somatosensory vs. visual as functional specialization index = $\max(S_i) - \text{mean}(S_i)$, and was normalized to 1 across 379 atlas partitions (Fig. S1).

## Brain asymmetry

Asymmetry maps were computed by contrasting intensity values in the right and left cortical hemispheres (RH-LH) in CIFTI space, independently for each metric (gFCD, myelin, cortical curvature, thickness, and sulcal depth maps) and individual.

## Interhemispheric vertex correspondence

Meaningful comparisons between the two hemispheres require high interhemispheric vertex correspondence[33]. We identified 32,492 homologous vertex pairs in the left and right hemispheres by determining their Cartesian coordinates relative to the bounding box center ($xyz = 180$ mm, 218 mm, 180 mm). Utilizing correlation analysis, we verified the interhemispheric correspondence of these coordinates, yielding a remarkable accuracy exceeding 99% ($R > 0.995$; Fig. S20).

## Brain activation patterns to hand movements

The Human Connectome Project (HCP) activation patterns to a motor task, which required subjects to tap their left or right fingers, or squeeze their left or right toes, or move their tongue[27,66], were used to determine homologous motor areas in the left and right brain hemispheres. Specifically, the HCP S1200 average task-fMRI Cohen's d effect-size maps across 997 subjects in CIFTI format were downloaded from the Brain Analysis Library of Spatial Maps and Atlases (https://balsa.wustl.edu/).

## Statistical analyses

Before vertex-wise statistical analysis, we removed site- and scanner-specific differences, regressed out FD across participants, and removed unwanted effects associated with age, sex, and race [White, African American, Hispanic, Asian, Other], independently for the L, R, and M groups in Matlab. Two-sample t-test (2-sided) was used to map gFCD/rsFC differences between left- and right-handers in Matlab. A false discovery rate (FDR) corrected threshold pFDR < 0.05 was used to correct for multiple comparisons across 91,282 gray ordinates in the CIFTI data. ROI analyses assessing group differences in diffusion MRI metrics and the reproducibility of group differences in gFCD and rsFC were conducted in R v4.0 using ANCOVA with 7 covariates age, sex, race, research site, and brain volume (myelin, cortical thickness, curvature, sulcal depth, FA, MD, lD, and tD), scanner manufacturer, and FD (gFCD and rsFC). A Bonferroni correction (2-sided) for 42 major white matter tracts in the AtlasTrack (https://www.nitrc.org/projects/atlastrack) was used to correct white matter diffusion results for multiple comparisons.

## Reporting summary

Further information on research design is available in the Nature Portfolio Reporting Summary linked to this article.

## Data availability

ABCD data are publicly available through the National Institute of Mental Health Data Archive (nda.nih.gov). The HCP S1200 average task-fMRI Cohen's d effect-size maps are available for download from the Brain Analysis Library of Spatial Maps and Atlases (https://balsa.wustl.edu/). Both Individual ROI and group-averaged imaging data generated in this study are provided in the Source Data file. Source data are provided in this paper.

## Code availability

The MATLAB code to compute the gFCD maps in this study has been deposited in the Figshare database under accession code 25314982[65].

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

## Acknowledgements
We are thankful to Adam Thomas, Ph.D., Dustin Moraczewski, Ph.D., and Eric Earl, BS (National Institute of Mental Health Data Science and Sharing Team) for providing access to the ABCD Community MRI Collection (NDA collection 3165) data on our servers. This study utilized the computational resources of the NIH HPC Beowulf cluster. (http://hpc.nih.gov). This work was done with support from the National Institute on Alcohol Abuse and Alcoholism (Y1AA-3009; ZIAAA000550).

Data used in the preparation of this article were obtained from the ABCD Study (https://abcdstudy.org/) and are held in the NIMH Data Archive. The ABCD Study is supported by the National Institutes of Health (NIH). ABCD consortium investigators did not participate in the analysis or writing of this report. This manuscript reflects the views of the authors and may not reflect the opinions or views of the NIH or ABCD consortium investigators.

## Author contributions
D.T. and N.D.V. designed the study, D.T. performed statistical analyses and D.T. and N.D.V. wrote the paper.

## Funding

## Competing interests
The authors declare no competing interests.
