## [Peer Review File · Nature Communications]

Associations between Handedness and Brain Functional Connectivity Patterns in ChildrenREVIEWER COMMENTS

Reviewer #1 (Remarks to the Author):

Tomasi and colleagues conducted a noteworthy study utilizing the ABCD cohort to investigate the relationship between handedness and global brain functional connectivity (FC) in children. Their findings indicated that left-handers exhibited greater FC in the left-hand motor cortex (located in the right hemisphere) and lower FC in the right-hand motor cortex (located in the left hemisphere). Additionally, they introduced a novel hand-motor composite index of handedness that effectively differentiated between right-handers and left-handers. It's worth noting that the association between handedness and motor cortex function appeared to be dependent on the imaging modality, with no statistically significant differences in structural measures, including morphological measures in the grey matter and structural connectivity in the white matter. This study is very interesting and demonstrates a valid experimental design with a reasonable selection of statistical models. Furthermore, the robustness of these crucial findings is reinforced by their ability to be replicated in an additional dataset. This not only enhances our understanding of the relationship between handedness and motor cortex function but also extends the generalizability of these findings to future research. I do, however, have a few comments and suggestions that may further enhance the current paper.

1. The influence of homotopic FC (i.e., "mirror FC"): In the "gFCD" section of the Methods, the authors computed the gFCD by counting the total number of edges between grayordinates throughout the entire brain. These edges were exclusively limited to a correlation coefficient greater than 0.6. Given that homotopic FC, particularly in the motor cortex, tends to be quite robust, it raises questions about whether these mirror FC connections could be driving the observed results, such as the increased FC in the left-hand motor cortex and reduced FC in the right-hand motor cortex in left-handers. It would be beneficial to investigate whether the results remain consistent when the authors exclude homotopic FC, especially in brain regions that exhibit an association with handedness. Alternatively, the gFCD map could be divided into intra-hemispheric gFCD and inter-hemispheric gFCD maps for a more detailed analysis.

2. In the "Global FCD" section of the Results, although the authors did not find any significant differences in morphological measures and white matter structural connectivity

between right- and left-handers, it would be beneficial to include unthresholded statistical maps in the supplementary information. These maps can visually represent brain regions that may have exhibited marginal significance in between-group differences, potentially due to sample size limitations. Providing this additional information can serve as a valuable resource for future research and offer insights into regions of interest for further investigation.

3. In the "Functional lateralization" section of the Results, the authors used rsFC(Mr) – rsFC(Ml) to quantify regional asymmetry. However, it may be worthwhile to consider normalizing the bilateral rsFC strength using the formula $(\text{right} - \text{left}) / (\text{right} + \text{left})$ to account for potential variations in the overall strength of connectivity. This normalization approach could provide a more balanced assessment of lateralization and may offer additional insights into the functional differences between right and left-handers.

4. In the "Effect of handedness on lateralization" section of the Results, the authors employed a vertex-wise analysis to assess whole-brain differences in the asymmetry index between left-handers and right-handers. It might be more informative to focus on identifying between-group differences in regions exhibiting significant asymmetry effects, rather than across the entire brain. I am curious about the authors' considerations regarding this approach and whether they explored analyzing specific regions of interest showing significant asymmetry effects.

5. Likely, in Fig. 6c, the authors identified significant between-group differences in gFCD asymmetry of the motor cortex. I am not sure whether this significant region was tested among the regions showing significant asymmetry effect on gFCD, or if it was assessed separately.

6. In the "gFCD" section of the Methods, the authors calculated the total number of edges (correlation > 0.6) between grayordinates in the whole brain, and only positive connectivity was considered in the gFCD calculation. It would be beneficial to provide an explanation for the exclusion of strong negative connectivity with correlations below -0.6, as this decision could impact the interpretation of the results and the overall understanding of the connectivity patterns.

7. In the "Brain structural asymmetry" section of the Results, the authors only investigated the differences in asymmetry indices of morphological and white matter measures between the right and left hemispheres. It would be valuable to clarify whether the authors also

conducted whole-brain vertex-wise analyses to assess the differences in these measures between left- and right-handers. If such analyses were not performed, it might be worth considering including them to provide a more comprehensive understanding of structural differences in relation to handedness.

Reviewer #2 (Remarks to the Author):

The conducted study investigates the differences in functional connectivity patterns between left- and right-handed children. The results indicate a considerable difference in the functional architecture of left- and right-handed children, which is quite interesting for researchers investigating laterality in motor processes and might also be of interest for developmental scientists. While I'm generally in favor of seeing a published version of this study, there are several comments and questions that should be addressed prior to further consideration.

>>> Major comments <<<

-> While the introduction gives a nice overview about differences in brain asymmetries in adults, there is only little information about the developmental aspects of brain asymmetries and handedness during childhood. Given that your study focuses on a cohort of 9-11 year olds, I would suggest adding information on brain asymmetries and handedness in children, the development of brain asymmetries & handedness, as well as explaining your choice of study sample in more detail.

-> Meaningful comparisons between the two hemispheres need the establishment of hemispheric homologues. This problem gave rise to parcellation schemes that are particularly designed with hemispheric asymmetries in mind see for instance the AICHA-atlas (Joliet et al., 2015) or the "homotopic local-global parcellation" based on the Yeo atlas (Yan et al., 2023) which was published recently. On the surface-level, studies like Gotts and colleagues (2013) established a homotopy between left and right vertices based on their geometrical distance. However, I'm uncertain how exactly you defined hemisphere-homologous pairs of elements (being it vertices, parcells, networks) to compute your lateralization index. Given the importance of this technicality, I suggest you elaborate your

procedure for aligning the hemispheres or defining hemispheric homologs in a dedicated section within your methods.

>>> Minor and detailed comments <<<

> Results <

line 77: "The prevalence of handedness for the ABCD cohort corresponded to 79.4% for right-handers, 7.14% for left-handers and 13.49% for mixed handers."

-> the number for mixed-handedness/ambidextrous participants is quite high in comparison to numbers reported in adult samples. Please elaborate on this divergence in the discussion section. Furthermore, please report the total numbers in addition to the percentages.

Demographics for the used samples

-> it does not become apparent that you use selected samples of right-, left-, and mixed-handed participants for your analyses. This could be clarified in the section "demographics for the ABCD 2.0 data release", to avoid potential confusion by the readers. Furthermore, the demographics of the exact samples (n=600 per sample) you use for statistical comparisons of the gFCD are missing. Adding this information would allow your study to be used for potential meta-analyses in the future, thus I highly recommend adding this information.

line 100: handedness index

-> please explain the used formula for calculating the handedness inventory. laterality indices are traditionally calculated with resemblance to the original Laterality Quotient by (Oldfield, 1971). I'm particularly uncertain about the multiplication by 2. Please elaborate.

line 118: "There were no significant effects of handedness on brain morphometrics (sulcal depth, cortical thickness and curvature), cortical myelin, or white matter diffusion metrics (fractional anisotropy, FA, mean, MD, longitudinal, ID, and transverse, tD, diffusivities)"

-> Please report the exact statistics in the supplementary material

line 122: "In right-handers, the Mr and MI seeds in Brodmann area 4 had strong average rsFC [...]"

-> here, I expect that the average rsFC is different from gFCD, but are not certain. Please elaborate. If average rsFC is simply the mean node-to-node rsFC, you could explain why exactly you didn't use gFCD for this comparison too. Otherwise if both wordings refer to the same metric, please consider rephrasing to avoid confusion.

line 146: figure 4 "[...] laterality of the functional connectivity [...]"

-> it is not directly clear to me what the contrasts images show: what exactly is meant with functional connectivity here? averaged rsFC, gFCD or the difference FCmaps when seeding from Mr or MI? Please clarify.

line 179: specialization

It is not clear to me what exactly is meant with functional specialization or if this is needed. I believe the message should be that differences between left and right handers (in functional connectivity, I assume) reside primarily in in homomodal areas that belong to primary or secondary functional cortices. If so, please consider introducing the reasoning behind this paragraph at its beginning (e.g.: "To characterize differences in DELTA between right- and left-handers in terms of the functional processing hierarchy, we computed the functional specialization index which differentiates between [...]")

line 214: "However, the asymmetry patterns of the MRI and white matter diffusion metrics did not reveal reproducible effects of handedness (data not shown)"

-> this feels a bit weird to report it and not show the data. Please reconsider as this might be of interest for other researchers who study structural variability.

> Discussion <

line 233: "These findings are also consistent with the differences in the neuroanatomy of handwriting between right-handers and converted left-handers"

-> What exactly is the finding you are referring to? Please elaborate.

line 256: "Heteromodal brain areas (24dd, POS2, FST, BA40, and IPS1) also demonstrated differential lateralization for right- and left-handers suggesting that handedness impact regions beyond the motor pathways"

-> Which metric is meant here? Please elaborate.

Line 267: "Here we also document reproducible group differences in cerebellar asymmetry, such that left- handers had weaker rightward lateralization in the left cerebellar hemisphere and stronger leftward lateralization in the right hemisphere than right-handers"

-> Again, I'm uncertain which metric is meant in this sentence. Please elaborate.

Line 274: "The asymmetries of the structural brain metrics (cortical myelin, thickness, curvature, sulcal depth, FA, MD, ID, and tD) were consistent across children with varying hand preferences and there were not differences in laterality between left- and right-handed children."

-> I'm uncertain about this statement, as you report that the data is not shown. Maybe you could analytically compare the distribution of structural asymmetry patterns for example by means of a dice coefficient or a direct contrast map between left- and right-handers

> Materials and Methods <

Line 313: "In the ABCD 2.0 data release³⁶ we identified a group of 600 left-handed children"

-> This reads a bit confusing, as it makes me wonder how you identified the handedness of the participants. It only becomes apparent later, that all children were classified using the EHIS. Please consider rephrasing.

Line 359: Table 1

-> I'm uncertain which table is meant. There seems to be no table 1 in the manuscript, and the supplementary table 1 shows something different.

Line 402: "Regions-of-interest (ROIs)"

-> I might have overlooked it but couldn't find the atlas/parcellation scheme that was used. Please include this information.

Line 417: "Asymmetry maps were computed by contrasting intensity values in the right and left cortical hemispheres (RH-LH) in CIFTI space, independently for each metric (gFCD, myelin, cortical curvature, thickness, and sulcal depth maps) and individual."

-> I'm wondering how exactly you ensured that both hemispheres were structurally homologous (see major comment).

Reviewer #1 (Remarks to the Author):

General comment: “Tomasi and colleagues conducted a noteworthy study utilizing the ABCD cohort to investigate the relationship between handedness and global brain functional connectivity (FC) in children. Their findings indicated that left-handers exhibited greater FC in the left-hand motor cortex (located in the right hemisphere) and lower FC in the right-hand motor cortex (located in the left hemisphere). Additionally, they introduced a novel hand-motor composite index of handedness that effectively differentiated between right-handers and left-handers. It's worth noting that the association between handedness and motor cortex function appeared to be dependent on the imaging modality, with no statistically significant differences in structural measures, including morphological measures in the grey matter and structural connectivity in the white matter. This study is very interesting and demonstrates a valid experimental design with a reasonable selection of statistical models. Furthermore, the robustness of these crucial findings is reinforced by their ability to be replicated in an additional dataset. This not only enhances our understanding of the relationship between handedness and motor cortex function but also extends the generalizability of these findings to future research”.

Response: Thank you for your thoughtful and encouraging comments. We appreciate your acknowledgment of the novelty of our study and the robustness of our experimental design.

Comment 1. “The influence of homotopic FC (i.e., ‘mirror FC’): In the ‘gFCD’ section of the Methods, the authors computed the gFCD by counting the total number of edges between grayordinates throughout the entire brain. These edges were exclusively limited to a correlation coefficient greater than 0.6. Given that homotopic FC, particularly in the motor cortex, tends to be quite robust, it raises questions about whether these mirror FC connections could be driving the observed results, such as the increased FC in the left-hand motor cortex and reduced FC in the right-hand motor cortex in left-handers. It would be beneficial to investigate whether the results remain consistent when the authors exclude homotopic FC, especially in brain regions that exhibit an association with handedness. Alternatively, the gFCD map could be divided into intra-hemispheric gFCD and inter-hemispheric gFCD maps for a more detailed analysis”.

Response: Thank you for your insightful comments and suggestions regarding the influence of homotopic functional connectivity (FC) in our study. To address your concerns, we conducted additional vertex-wise analyses dividing the gFCD map into intra-hemispheric and inter-hemispheric components as you suggested. The patterns of inter-hemispheric gFCD did not differ between L and R in any brain region, and the corresponding Handedness index did not differentiate the handedness subgroups suggesting that homotopic connectivity did not drive the group differences in gFCD in the hand-motor cortex. These results are now highlighted in the new Results section “Ipsilateral and contralateral gFCD”.

Comment 2. “In the ‘Global FCD’ section of the Results, although the authors did not find any significant differences in morphological measures and white matter structural connectivity between right- and left-handers, it would be beneficial to include unthresholded statistical maps in the supplementary information. These maps can visually represent brain regions that may have exhibited marginal significance in between-group differences, potentially due to sample size limitations. Providing this additional information can serve as a valuable resource for future research and offer insights into regions of interest for further investigation.”

Response: We appreciate your attention to detail and the opportunity to enhance the comprehensiveness of our results. The revised version of the supplementary information now includes unthresholded statistical maps as you recommended.

Comment 3. "In the 'Functional lateralization' section of the Results, the authors used rsFC(Mr) – rsFC(Ml) to quantify regional asymmetry. However, it may be worthwhile to consider normalizing the bilateral rsFC strength using the formula $(\text{right} - \text{left}) / (\text{right} + \text{left})$ to account for potential variations in the overall strength of connectivity. This normalization approach could provide a more balanced assessment of lateralization and may offer additional insights into the functional differences between right and left-handers."

Response: We appreciate the insightful suggestion from the reviewer regarding the quantification of the "Functional lateralization". In the revised version we now assess the effect of handedness on the lateralization of hand-motor connectivity using the suggested normalization method. We sincerely thank the reviewer for his/her thoughtful input, which we believe contributes to the overall quality and precision of our study.

Comment 4. "In the 'Effect of handedness on lateralization' section of the Results, the authors employed a vertex-wise analysis to assess whole-brain differences in the asymmetry index between left-handers and right-handers. It might be more informative to focus on identifying between-group differences in regions exhibiting significant asymmetry effects, rather than across the entire brain. I am curious about the authors' considerations regarding this approach and whether they explored analyzing specific regions of interest showing significant asymmetry effects."

Response: In this section, we used the vertex-wise analysis to identify regions with significant group differences in the laterality index. The opening sentence of this section now states the purpose of the whole-brain analysis.

Comment 5. "Likely, in Fig. 6c, the authors identified significant between-group differences in gFCD asymmetry of the motor cortex. I am not sure whether this significant region was tested among the regions showing significant asymmetry effect on gFCD, or if it was assessed separately."

Response: The reviewer is correct. The significant differences in gFCD asymmetry encompassed the entire Mr ROI as illustrated in Fig 7c. The last sentence in the gFCD asymmetry section now highlights this.

Comment 6. "In the 'gFCD' section of the Methods, the authors calculated the total number of edges (correlation > 0.6) between grayordinates in the whole brain, and only positive connectivity was considered in the gFCD calculation. It would be beneficial to provide an explanation for the exclusion of strong negative connectivity with correlations below -0.6, as this decision could impact the interpretation of the results and the overall understanding of the connectivity patterns."

Response: to address the reviewer's concern we additionally computed negative gFCD using the correlation threshold < -0.6. This analysis did not reveal group differences in negative gFCD. Thus the effect of handedness on gFCD is not confounded by negative edges. The opening sentence of "Global FCD" section in Results now states that we used the standard correlation threshold 0.6 to compute gFCD and the closing sentence highlight the lack of significant effects of handedness on negative gFCD.

Comment 7. "In the 'Brain structural asymmetry' section of the Results, the authors only investigated the differences in asymmetry indices of morphological and white matter measures between the right and left hemispheres. It would be valuable to clarify whether the authors also conducted whole-brain vertex-wise analyses to assess the differences in these measures between left- and right-handers. If such analyses were not performed, it might be worth considering including them to provide a more comprehensive understanding of structural differences in relation to handedness."

Response: the reviewer is correct; we conducted vertex-wise statistical analysis to assess differences in structural asymmetry between left- and right-handers. However, the patterns of structural asymmetry did not reveal significant effects of handedness. These results are now reported in the Supplement (Fig S18).

Reviewer #2 (Remarks to the Author):

The conducted study investigates the differences in functional connectivity patterns between left- and right-handed children. The results indicate a considerable difference in the functional architecture of left- and right-handed children, which is quite interesting for researchers investigating laterality in motor processes and might also be of interest for developmental scientists.

Response: Thank you for your thoughtful and positive feedback on our study. We appreciate your acknowledgment of the significance of the research.

Comment1: While the introduction gives a nice overview about differences in brain asymmetries in adults, there is only little information about the developmental aspects of brain asymmetries and handedness during childhood. Given that your study focuses on a cohort of 9-11 year olds, I would suggest adding information on brain asymmetries and handedness in children, the development of brain asymmetries & handedness, as well as explaining your choice of study sample in more detail.

Response: In the Introduction we now provide information on handedness in children and summarize the few studies on development of brain asymmetries and its linkage with handedness. In the Introduction we also explain in greater detail the choice of the study sample.

Comment2: Meaningful comparisons between the two hemispheres need the establishment of hemispheric homologues. This problem gave rise to parcellation schemes that are particularly designed with hemispheric asymmetries in mind see for instance the AICHA-atlas (Joliet et al., 2015) or the "homotopic local-global parcellation" based on the Yeo atlas (Yan et al., 2023) which was published recently. On the surface-level, studies like Gotts and colleagues (2013) established a homotopy between left and right vertices based on their geometrical distance. However, I'm uncertain how exactly you defined hemisphere-homologous pairs of elements (being it vertices, parcels, networks) to compute your lateralization index. Given the importance of this technicality, I suggest you elaborate your procedure for aligning the hemispheres or defining hemispheric homologs in a dedicated section within your methods.

Response: Thank you for your insightful comments and suggestions on our manuscript. We agree on the importance of establishing hemispheric homologues for meaningful comparisons between the two hemispheres. To identify 32,492 homologous vertex pairs in the left and right hemispheres we first determined their Cartesian coordinates, relative to the center of the image volume bounding box (xyz = 180mm, 218mm, 180mm) and then used correlation analysis to check the interhemispheric

correspondence of the Cartesian coordinates (Fig SX). We incorporated a dedicated section within the revised Methods that elaborates on our procedure for identifying homologous pairs.

Comment3: line 77: "The prevalence of handedness for the ABCD cohort corresponded to 79.4% for right-handers, 7.14% for left-handers and 13.49% for mixed handers. The number for mixed-handedness/ambidextrous participants is quite high in comparison to numbers reported in adult samples. Please elaborate on this divergence in the discussion section. Furthermore, please report the total numbers in addition to the percentages. "

Response: The reviewer is correct. The percentage of left-handed children (7.14%) was lower, while the proportion of mixed-handed individuals (13.49%) was higher compared to figures reported in studies involving adults (9.33% for left-handedness and 10.6% for mixed-handedness). The limitation paragraph at the end of Discussion now interprets this as suggesting that at baseline, ABCD children may not have fully established their handedness, and some may still be in the process of developing a dominant hand.

Comment4: "It does not become apparent that you use selected samples of right-, left-, and mixed-handed participants for your analyses. This could be clarified in the section "demographics for the ABCD 2.0 data release", to avoid potential confusion by the readers. Furthermore, the demographics of the exact samples (n=600 per sample) you use for statistical comparisons of the gFCD are missing. Adding this information would allow your study to be used for potential meta-analyses in the future, thus I highly recommend adding this information."

Response: Thank you for your suggestions. Results now alert readers that we selected 600 children in each subgroup and report the related demographics in Table 1, which can be found after the Reference list.

Comment5: "line 100: handedness index-> please explain the used formula for calculating the handedness inventory. laterality indices are traditionally calculated with resemblance to the original Laterality Quotient by (Oldfield, 1971). I'm particularly uncertain about the multiplication by 2. Please elaborate."

Response: We thanks the reviewer for noticing this mistake in the formula which is corrected in the revised manuscript. The revised Discussion now highlights that our neuroimaging index of handedness serves as a neuroimaging equivalent to Oldfield's laterality quotient.

Comment6: "line 118: 'There were no significant effects of handedness on brain morphometrics (sulcal depth, cortical thickness and curvature), cortical myelin, or white matter diffusion metrics (fractional anisotropy, FA, mean, MD, longitudinal, ID, and transverse, tD, diffusivities). Please report the exact statistics in the supplementary material"

Response: We now present the exact statistics in the supplementary material as requested.

Comment7: "line 122: 'In right-handers, the Mr and MI seeds in Brodmann area 4 had strong average rsFC [...] here, I expect that the average rsFC is different from gFCD, but are not certain. Please elaborate. If average rsFC is simply the mean node-to-node rsFC, you could explain why exactly you didn't use gFCD for this comparison too. Otherwise if both wordings refer to the same metric, please

consider rephrasing to avoid confusion.”

Response: We apologize for the lack of clarity. We used gFCD to assess the number of positive edges of each vertex in the dense connectome, and gFCD-guided seed-voxel correlation analyses to explore the functional connectivity patterns of gFCD clusters (MI and Mr). We added opening sentences to the “Global FCD” and “Connectivity of the hand motor area” Method sections to avoid confusion.

Comment8: “line 146: figure 4 ‘[...] laterality of the functional connectivity [...]’ it is not directly clear to me what the contrasts images show: what exactly is meant with functional connectivity here? averaged rsFC, gFCD or the difference FCmaps when seeding from Mr or MI? Please clarify.”

Response: We apologize for the lack of clarity. Figure 4 shows differences in the laterality index, which is based on seed-voxel correlation metrics. This is now clarified in the figure legend.

Comment9: “line 179: specialization. It is not clear to me what exactly is meant with functional specialization or if this is needed. I believe the message should be that differences between left and right handers (in functional connectivity, I assume) reside primarily in in homomodal areas that belong to primary or secondary functional cortices. If so, please consider introducing the reasoning behind this paragraph at its beginning (e.g.: ‘To characterize differences in DELTA between right- and left-handers in terms of the functional processing hierarchy, we computed the functional specialization index which differentiates between [...]’)”

Response: We added an opening sentence to the section as suggested.

Comment10: “line 214: ‘However, the asymmetry patterns of the MRI and white matter diffusion metrics did not reveal reproducible effects of handedness (data not shown)’ This feels a bit weird to report it and not show the data. Please reconsider as this might be of interest for other researchers who study structural variability.”

Response: We now show the data using uncorrected statistics in Supplementary material as suggested.

Comment11: “line 233: ‘These findings are also consistent with the differences in the neuroanatomy of handwriting between right-handers and converted left-handers’ What exactly is the finding you are referring to? Please elaborate.”

Response: We apologize for the lack of clarity. The sentence was reworded as “These findings are also consistent with the differences in cerebral blood flow changes induced by handwriting between right-handers and converted left-handers.”

Comment12: “line 256: ‘Heteromodal brain areas (24dd, POS2, FST, BA40, and IPS1) also demonstrated differential lateralization for right- and left-handers suggesting that handedness impact regions beyond the motor pathways/ Which metric is meant here? Please elaborate.”

Response: We apologize for the lack of clarity. The sentence referred to the lateralization index reflecting differential connectivity with the left- and right-hand motor areas. The sentence has been reworded accordingly.

Comment13: “Line 267: ‘Here we also document reproducible group differences in cerebellar

asymmetry, such that left-handers had weaker rightward lateralization in the left cerebellar hemisphere and stronger leftward lateralization in the right hemisphere than right-handers' Again, I'm uncertain which metric is meant in this sentence. Please elaborate."

Response: We apologize for the lack of clarity. The sentence referred to the lateralization index reflecting differential connectivity with the left- and right-hand motor areas. The sentence has been reworded accordingly.

Comment14: "Line 274: 'The asymmetries of the structural brain metrics (cortical myelin, thickness, curvature, sulcal depth, FA, MD, ID, and tD) were consistent across children with varying hand preferences, and there were not differences in laterality between left- and right-handed children.' I'm uncertain about this statement, as you report that the data is not shown. Maybe you could analytically compare the distribution of structural asymmetry patterns for example by means of a dice coefficient or a direct contrast map between left- and right-handers"

Response: We added Dice coefficients to the Results to support our interpretations.

Comment15: "Line 313: 'In the ABCD 2.0 data release³⁶ we identified a group of 600 left-handed children' This reads a bit confusing, as it makes me wonder how you identified the handedness of the participants. It only becomes apparent later, that all children were classified using the EHIS. Please consider rephrasing."

Response: The sentence has been reworded to clarify that EHIS was used to identify the group of 600 left-handed children.

Comment16: "Line 359: Table 1. I'm uncertain which table is meant. There seems to be no table 1 in the manuscript, and the supplementary table 1 shows something different.

Response: We believe Table 1 was included in the original submission after the Reference list, and sincerely apologize if this was not the case. Please find Table 1 following the Reference list.

Comment17: Line 402: "Regions-of-interest (ROIs)" I might have overlooked it but couldn't find the atlas/parcellation scheme that was used. Please include this information.

Response: The Supplementary Table S1 contains the ROI centroid locations, and the statistics of the laterality of the functional connectivity with the hand motor cortex.

Comment18: Line 417: "Asymmetry maps were computed by contrasting intensity values in the right and left cortical hemispheres (RH-LH) in CIFTI space, independently for each metric (gFCD, myelin, cortical curvature, thickness, and sulcal depth maps) and individual." I'm wondering how exactly you ensured that both hemispheres were structurally homologous (see major comment).

Response: To identify 32,492 homologous vertex pairs in the left and right hemispheres we first determined their Cartesian coordinates, relative to the center of the image volume bounding box (xyz = 180mm, 218mm, 180mm) and then used correlation analysis to check the interhemispheric correspondence of the Cartesian coordinates X, Y, Z. We incorporated a dedicated section within the revised Methods that elaborates on our procedure for identifying homologous pairs.

REVIEWERS' COMMENTS

Reviewer #1 (Remarks to the Author):

The authors addressed all of my concerns. Well done!! It's a very interesting study. I prefer to highlight this study in Nature Communications.

Reviewer #2 (Remarks to the Author):

Thank you for the productive revision process! My sincere apologies for the delay in response.

All of my comments have been addressed in a satisfying manner. Therefore I'm looking forward to seeing a published version of this manuscript.